# The structures of two archaeal type IV pili illuminate evolutionary relationships

Fengbin Wang[1,5], Diana P. Baquero [2,3,5], Zhangli Su[1], Leticia C. Beltran[1], David Prangishvili[2,4], Mart Krupovic[2 ✉] & Edward H. Egelman [1✉]

We have determined the cryo-electron microscopic (cryo-EM) structures of two archaeal type IV pili (T4P), from *Pyrobaculum arsenaticum* and *Saccharolobus solfataricus*, at 3.8 Å and 3.4 Å resolution, respectively. This triples the number of high resolution archaeal T4P structures, and allows us to pinpoint the evolutionary divergence of bacterial T4P, archaeal T4P and archaeal flagellar filaments. We suggest that extensive glycosylation previously observed in T4P of *Sulfolobus islandicus* is a response to an acidic environment, as at even higher temperatures in a neutral environment much less glycosylation is present for *Pyrobaculum* than for *Sulfolobus* and *Saccharolobus* pili. Consequently, the *Pyrobaculum* filaments do not display the remarkable stability of the *Sulfolobus* filaments in vitro. We identify the *Saccharolobus* and *Pyrobaculum* T4P as host receptors recognized by rudivirus SSRV1 and tristromavirus PFV2, respectively. Our results illuminate the evolutionary relationships among bacterial and archaeal T4P filaments and provide insights into archaeal virus-host interactions.

[1] Department of Biochemistry and Molecular Genetics, University of Virginia, Charlottesville, VA 22908, USA. [2] Archaeal Virology Unit, Department of Microbiology Institut Pasteur, 25 rue du Dr. Roux, Paris 75015, France. [3] Sorbonne Universités, Collège Doctoral, Paris 75005, France. [4] Ivane Javakhishvili Tbilisi State University, Tbilisi 0179, Georgia. [5]These authors contributed equally: Fengbin Wang, Diana P. Baquero. ✉email: mart.krupovic@pasteur.fr; egelman@virginia.edu

It has been recognized for many years that bacterial type IV pili (T4P)[1,2], important in activities such as adhesion[3,4], twitching motility[5], surface-sensing[6] and natural transformation[7], are homologs of two different classes of archaeal surface structure, pili and flagellar filaments[8–11]. The archaeal flagellar filament is a unique rotating T4P-like assembly which enables motility and is conserved in widely different archaeal lineages, including hyperthermophiles, halophiles and methanogens[12]. Archaeal T4P are even more widespread and functionally diverse[13]. Although the full spectrum of functions played by archaeal T4P is yet to be uncovered, they are known to mediate adhesion to various biotic and abiotic surfaces, and play an important role in DNA exchange, intercellular communication and biofilm formation[14,15]. Furthermore, T4P serve as receptors for certain hyperthermophilic archaeal viruses, including Sulfolobus islandicus rod-shaped viruses 2 and 8 (SIRV2 and SIRV8, respectively; family *Rudiviridae*)[16,17] and Sulfolobus turreted icosahedral virus (STIV; family *Turriviridae*)[18].

Despite the importance of bacterial and archaeal T4P-like filaments, the structural details of these filaments remain sparse. However, the revolution in cryo-EM has now led to atomic structures for seven bacterial T4P[19–21], three archaeal flagellar filaments[22–24], one archaeal "flagellar-like" filament[25], and one archaeal T4P[26]. Interestingly, a 4.1-Å resolution structure of the T4P from *S. islandicus*, a hyperthermophilic and acidophilic crenarchaeon, revealed an unprecedented level of O-glycosylation on this filament[26]. The adhesive pilus displayed an anomalous amino acid composition, with 37% of the residues in the C-terminal globular domain being threonines and serines, most of which are exposed and likely glycosylated on the surface of the filament. The core of the filament was, as expected, extremely hydrophobic, since the N-terminal domains that constitute the core are transmembrane helices prior to filament assembly. However, the overall percentage of charged residues within the entire pilin protein was only 1.5%. Remarkably, the *S. islandicus* pilus was found to be extremely resilient under various harsh conditions, including extended treatment with different proteases and even boiling in 5 M guanidine hydrochloride. The reason for the robustness of the pilus was attributed to a combination of profound hydrophobicity and extensive surface glycosylation of the filament[26]. Notably, the resilience of the *S. islandicus* pilus precluded the identification of the constituent pilin protein using conventional approaches, such as gel electrophoresis and mass spectrometry. Instead, the identity of the protein was determined using a combination of cryo-EM and bioinformatics, whereby candidate protein sequences were directly threaded through the density maps, which allowed building an atomic model de novo, exemplifying the power of cryo-EM for protein identification[26].

Whether T4P of other hyperthermophilic archaea display similar sequence features and properties as those described for *S. islandicus* pilus remains unclear. Indeed, the existence of a single atomic structure for an archaeal T4P has limited generalizations about such filaments as well as comparisons with homologs. We present in this paper the structure of two additional archaeal T4P, one from *Pyrobaculum arsenaticum*, a neutrophilic hyperthermophile (optimal growth at 90 °C, pH 7) and the other from *Saccharolobus solfataricus*, an acidophilic hyperthermophile (optimal growth at 75–80 °C, pH 2-3). We use the three structures of archaeal T4P to make comparisons with bacterial T4P and archaeal flagellar filaments. We show that whereas the N-terminal domain is conserved in all bacterial and archaeal T4P-like filaments, the C-terminal domain of bacterial T4P is unrelated to the immunoglobulin (Ig)-like domain shared by archaeal T4P and archaeal flagella. Furthermore, comparison of the pili from the acidophilic and neutrophilic hyperthermophiles suggests that extensive glycosylation is a response to an acidic

environment, rather than simply high temperature. Notably, the pili of *Pyrobaculum* are much less glycosylated compared to the *Sulfolobus* and *Saccharolobus* pili, and as a consequence, the *Pyrobaculum* filaments do not display the remarkable stability of the *Sulfolobus* filaments in vitro. Finally, we show that *Pyrobaculum* and the *Saccharolobus* T4P are at the forefront of virus-host interaction and are recognized by two different viruses, likely serving as primary receptors during virus infection.

## Results

**Archael viruses bind to pili.** We have recently isolated two filamentous viruses, Saccharolobus solfataricus rod-shaped virus 1 (SSRV1; family *Rudiviridae*) and Pyrobaculum filamentous virus 2 (PFV2; family *Tristromaviridae*), infecting *Saccharolobus solfataricus* POZ149 and *Pyrobaculum arsenaticum* 2GA, respectively[27,28]. After centrifugation in CsCl density gradient, both viruses co-purify with and are often bound with their terminal fibers to host-derived pili-like filaments, which were ~80 Å in diameter (Fig. 1a, b). Consequently, the filaments appear to serve as the primary receptors for SSRV1 and PFV2, as has been demonstrated for some other archaeal viruses[16–18].

**Cryo-EM structures of two archaeal pili.** In the absence of genetic tools for *S. solfataricus* POZ149 and *P. arsenaticum* 2GA, we set out to determine the identity of the SSRV1 and PFV2 host receptors directly by cryo-EM, applying the strategy recently employed for identification of the *S. islandicus* pilin[26]. To this end, mixtures of virions and pili-like filaments were imaged by cryo-EM separately for SSRV1 and PFV2, respectively, and the images of the pili were used for three-dimensional reconstruction. While multiple helical symmetries appeared to be possible based upon analysis of power spectra from these filaments[29], in each case only one helical symmetry led to a reconstruction with clearly interpretable secondary structure. This was a rise of ~5.3 Å and a rotation of ~101.7° for the *P. arsenaticum* filaments and a rise of ~5.0 Å and a rotation of ~104.6° for the *S. solfataricus* filaments (Table 1). The small difference in helical twist between the two resulted in 7-start long-pitch helices that were left-handed for *P. arsenaticum* (Fig. 1e, i) and right-handed for *S. solfataricus* (Fig. 1f, j). The 7-start helices arise from the fact that there are ~3.5 subunits per turn of the right-handed 1-start helix in both filaments. However, in *P. arsenaticum* pilus, there are slightly more than 3.5 subunits per turn, so there are 7.1 subunits in two turns, while in *S. solfataricus* there are slightly less than 3.5 subunits per turn, yielding 6.9 subunits in two turns. The reconstructions had a resolution of 3.8 Å for *P. arsenaticum* pilus and 3.4 Å for *S. solfataricus* pilus, as judged by a map:map Fourier shell correlation (FSC), a map:model FSC as well as $d_{99}$[30] (Supplementary Fig. 1, Table 1).

Hyperthermophilic archaea typically carry multiple pilin genes[13]. Thus, to identify the exact pilin genes encoding receptors of SSRV1 and PFV2, we have sequenced the complete genomes of *S. solfataricus* POZ149 and *P. arsenaticum* 2GA (see Methods), and used a combination of mass spectrometry, bioinformatics, and cryo-EM to build de novo the atomic models (Fig. 2a, b), as was done for the pili from *S. islandicus*[26]. The result in each case was finding a single protein sequence that could be threaded through the maps, which is HC235_07175 for *P. arsenaticum* (Par_PilA) and HFC64_06120 for *S. solfataricus* (Sso_PilA) (Supplementary Figs. 2, 3). Our structural models indicate that the first 12 residues in Sso_PilA and 7 residues in Par_PilA correspond to the signal sequence, which is cleaved from the mature pilin protein. In Sso_PilA, cleavage occurs between the Gly12 and Leu13 residues (Supplementary Fig. 2d) and the cleavage site matches perfectly with the previously established[11]

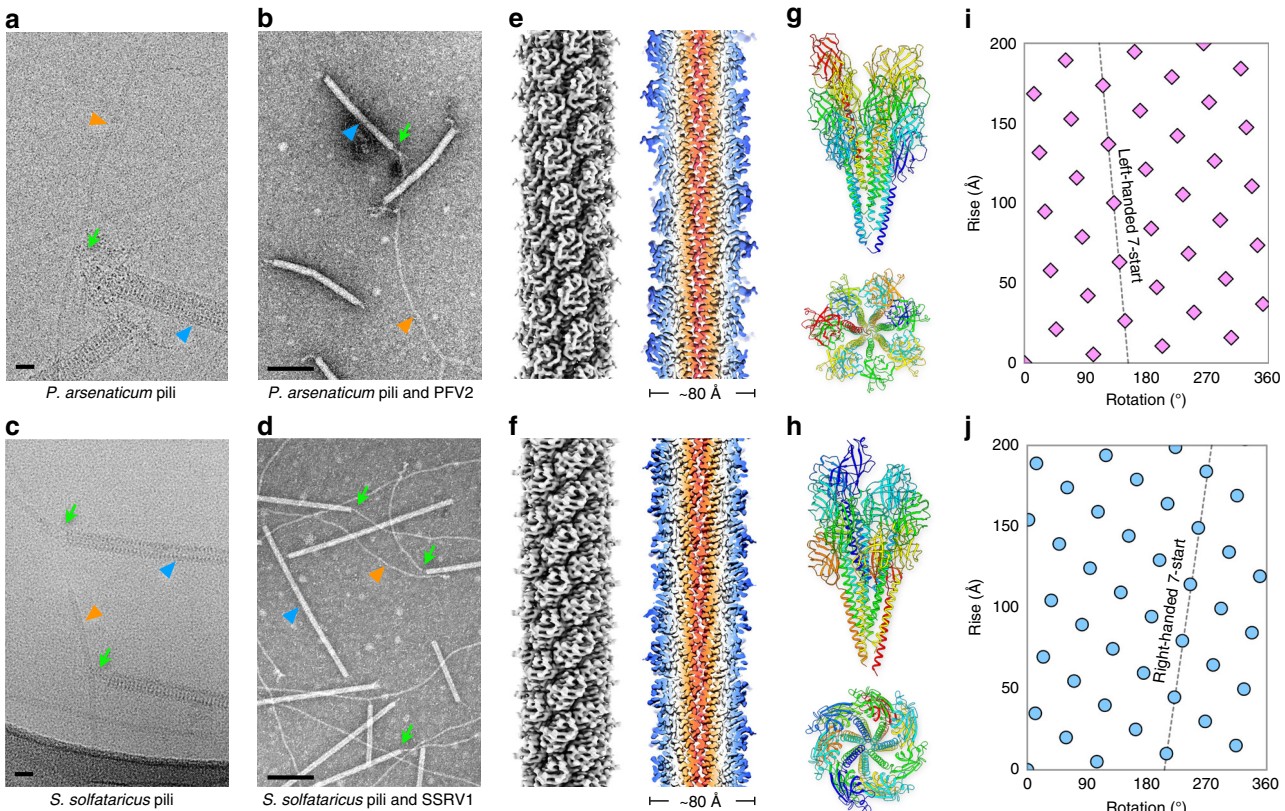

**Fig. 1 Cryo-EM of the *P. arsenaticum* pilus and the *S. solfataricus* pilus.** Representative cryo-electron (**a**) and negative staining (**b**) micrographs of the *P. arsenaticum* pili interacting with filamentous virus PFV2. Scale bar, 20 nm in **a** and 200 nm in **b**. Orange arrowhead points to the *P. arsenaticum* pilus; blue arrowhead points to the PFV2 virion; green arrow points to the regions of the pilus-virion interaction. Representative cryo-electron (**c**) and negative staining (**d**) micrographs of the *S. solfataricus* pili interacted with filamentous virus SSRV1. Scale bar, 20 nm in **c** and 200 nm in **d**. Orange arrowhead points to the *S. solfataricus* pilus; blue arrowhead points to the SSRV1 virion; green arrow points to the regions of the pilus-virion interaction. Cryo-EM reconstruction of the *P. arsenaticum* pilus at 3.8 Å resolution (**e**) and the *S. solfataricus* pilus at 3.4 Å resolution (**f**). Thin slices parallel to the helical axis of the pilus are shown, colored by the helical radius. **g**, **h** Side view and top view of the *P. arsenaticum* and *S. solfataricus* pilus atomic models, built into the cryo-EM maps shown in **e**, **f**. The model is colored by chain. Helical net of the pilus (*P. arsenaticum* pilus in **i** and the *S. solfataricus* pilus in **j**) using the convention that the surface is unrolled and we are viewing from the outside.

consensus recognition motif of the signal peptidase PibD ((Lys/Arg)-(Gly/Ala)-↓-(Leu/Ile/Phe)-(Ser/Thr/Ala)). In contrast, in Par_PilA, cleavage occurs between residues Gly7 and Met8 of the prepilin, and the PibD cleavage site (Arg-Gly↓Met-Thr) slightly deviates from the consensus motif (Supplementary Fig. 2c). BLASTP searches have shown that orthologs of Sso_PilA are conserved in other members of the order Sulfolobales, including different species of the genera *Saccharolobus*, *Sulfolobus*, *Acidianus* and others. Similarly, Par_PilA pilins are conserved in *Pyrobaculum* species. Notably, in a number of *Pyrobaculum* genomes, the orthologs of Par_PilA appear to be misannotated, with the second Met codon (corresponding to Met8 of Par_PilA prepilin) selected as the start codon; as a result, these proteins appear to lack the signal sequence.

The *S. solfataricus* pilus was very similar to that of *S. islandicus*, albeit at considerably higher resolution (3.4 versus 4.1 Å), which was not surprising given the ~81% sequence identity between the two. In contrast, the sequence identity between the pilins from *P. arsenaticum* and *S. solfataricus* is only ~18% (Fig. 2c). The structural comparison between the two pilins (Fig. 2d) reveals that, even with low sequence identity, both the N-terminal and C-terminal domains are structurally conserved. They both have the archaeal T4P two-domain fold described for the *S. islandicus* LAL14/1 pilus[26], containing a ~35 amino acid-long N-terminal α-helical domain and C-terminal globular Ig-like β-sandwich

domain. In fact, there is nearly a one-to-one mapping of secondary structural elements between the two (Fig. 2e), with the exception that one β-strand in *P. arsenaticum* (residues 36–46) maps to two shorter β-strands in *S. solfataricus*. However, the relative orientation of these two domains is different by ~40° (Fig. 2d) and, as a result, the helical packing in the filaments is different (Fig. 1g, h). Interestingly, and in contrast to the *S. islandicus* and *S. solfataricus* pilins, *P. arsenaticum* pilin has a 15 amino acid C-terminal extension that wraps around the globular domain. Within this extension, an intramolecular disulfide bond is formed between Cys139 in the extension and Cys80 within the same pilin chain (Fig. 2f, Supplementary Fig. 4), which is likely to contribute to the thermo-stability of this pilus.

**Glycosylation of the pili.** Despite high sequence similarity between the pilins of *S. islandicus* and *S. solfataricus*, SSRV1 does not bind to *S. islandicus* pili and, accordingly, cannot infect this organism[27]. Given that the surface of the *S. islandicus* pilus is coated with a glycan layer[26], the latter might play an important, if not key, role in virus binding. Thus, we next analyzed the glycosylation status of the *S. solfataricus* and *P. arsenaticum* pili. It turned out that amino acid distributions and glycosylation represent the most striking differences between the two pilins (Supplementary Table 1). In the *S. solfataricus* pilin, similar to *S.*

**Table 1 Cryo-EM and refinement statistics of the *P. arsenaticum* and the *S. solfataricus* pilus.**

| Parameter | *P. arsenaticum* pilus | *S. solfataricus* pilus |
|---|---|---|
| Data collection and processing | | |
| Voltage (kV) | 300 | 300 |
| Electron exposure (e⁻ Å⁻²) | 50 | 58 |
| Pixel size (Å) | 1.08 | 1.08 |
| Final particle images (n) | 210,331 | 579,713 |
| Helical symmetry | | |
| Point group | C1 | C1 |
| Helical rise (Å) | 5.26 | 4.97 |
| Helical twist (°) | 101.69 | 104.57 |
| Map resolution (Å) | | |
| Map:map FSC (0.143) | 3.8 | 3.4 |
| Model:map FSC (0.38) | 3.8 | 3.4 |
| $d_{99}$ | 3.9 | 3.6 |
| Refinement and model validation | | |
| Map-sharpening B-factor (Å²) | −140 | −104 |
| Bond lengths RMSD (Å) | 0.005 | 0.006 |
| Bond angles RMSD (°) | 1.004 | 0.986 |
| Model:map RSCC | 0.87 | 0.85 |
| Clashscore | 8.82 | 7.28 |
| Poor rotamers (%) | 0.5 | 0 |
| Ramachandran favored (%) | 95.63 | 95.73 |
| Ramachandran outlier (%) | 0 | 0 |
| MolProbity score | 1.78 | 1.70 |
| Deposition ID | | |
| PDB (model) | 6W8U | 6W8X |
| EMDB (map) | EMD-21578 | EMD-21579 |

*islandicus*, only about 1.5% of the residues are charged. We showed[26] that extensive glycosylation makes these extremely hydrophobic filaments soluble. In contrast, ~10% of the residues in *P. arsenaticum* pilin are charged. In *S. solfataricus* and *S. islandicus*, ~37% of the residues in the globular domain are either serine or threonine, potential sites for O-linked glycosylation. In *P. arsenaticum* pilin as well as archaeal flagella and adhesion filaments[22–26], only ~20% of the residues are serine or threonine (Fig. 3a). No extra density corresponding to O-linked glycosylation was reported, except for one serine in the flagellar filament of *M. hungatei*[22]. In *S. solfataricus*, we see clear extra density consistent with glycosylation on seven residues, all serines or threonines (Fig. 3b). In contrast, in *P. arsenaticum* only a single density consistent with glycosylation is seen, on Asn55 (Fig. 3b), which would be N-linked glycosylation.

We expect that most glycosylation is not seen in the high-resolution maps due to the disorder of the sugars, and the fact that the same residue may not be glycosylated in every subunit. However, we can directly visualize at lower resolution additional mass consistent with glycosylation (Fig. 3c) by subtracting the predicted density due to the atomic model from the actual maps, after both are filtered to 7 Å. We are fortunate to have a control for this procedure which involves the rod-shaped virus SSRV1 present in the images for the *S. solfataricus* pili. There was no apparent glycosylation of this virus, and the small extra density not explained by the atomic model involves a few disordered C-terminal residues of the capsid protein that were not part of the atomic model. It can be seen (Fig. 3c) that, as expected, the *S. solfataricus* filament has a significantly greater amount of peripheral density due to glycosylation than the *P. arsenaticum* filament. A prediction of the lesser glycosylation for the *P. arsenaticum* pilus is that it will not display the remarkable stability of the *S. islandicus* pilus in vitro[26]. Consistent with this prediction, boiling the *S. islandicus* pili in 5 M GuHCl failed to disassemble the filaments[26], but under the same conditions the *P. arsenaticum* pili come apart (Supplementary Fig. 5). Collectively, these results suggest that SSRV1 and PFV2 recognize their corresponding receptors through different interactions. Whereas binding of SSRV1 to *S. solfataricus* pili is likely to be mediated

through carbohydrate moieties, direct interaction with the pilin subunits might play an important role in the case of PFV2.

**Evolutionary relationships.** Expanding the number of archaeal T4P atomic structures from one to three allows for more reliable comparisons to now be made between these structures and both bacterial T4P and archaeal flagellar filaments. The most striking difference is that in both bacterial T4P and archaeal flagellins, the N-terminal α-helix is ~50 residues long, while in the archaeal T4P it is only ~35 residues long (Fig. 4a). By this criterion, the Iho670 adhesion filaments, previously described as "flagellar-like"[25], appear to be a form of archaeal T4P. Despite the differences in the N-terminal helix length between archaeal flagellins and T4P, the helical symmetries of the filaments are quite similar (Fig. 4b). In contrast, although bacterial T4P can have a similar twist to the archaeal filaments (~100°), they have a distinctly different axial rise of ~10 Å as opposed to the ~5 Å in the archaeal filaments. In addition, all known bacterial T4P show a partial melting of the N-terminal helix, not seen in any of the archaeal filaments[31].

The distribution of helical symmetries among the archaeal filaments and bacterial T4P also suggests a conservation of how their long N-terminal α-helical tails pack together (Fig. 4b). Therefore, five adjacent copies of those α-helical tails were cut out from the filament model and compared with each other (Fig. 4c, d). Interestingly, Iho670 adhesion filament packs very similarly to *S. solfataricus* and *S. islandicus* pili, with an RMSD < 2 Å, while *P. arsenaticum* packs slightly differently from those three filaments with RMSDs of 3–4 Å (Fig. 4d). Notably, the crucial N-terminal subunit-subunit interactions in these four filaments are almost all hydrophobic (Fig. 4a, c), except for in *P. arsenaticum* there is a glutamic acid, which may contribute to the slightly different packing compared to Iho670 and other archaeal T4P. The archaeal flagellar filaments, judging by the sequence alignment and the RMSD matrix (Fig. 4a, d), have a very similar packing with Iho670 and the other archaeal T4Ps in their first 35 amino acids. However, their helices extend further with more hydrophilic residues interacting both with each other and with the globular domains, presumably providing greater mechanical stability in their roles as rotating propellers. Bacterial T4Ps, in contrast, have more hydrophilic residues in their N-terminal helices, including the conserved glutamic acid-forming key hydrogen bonds with N-terminal residues. Because of the melted N-terminal region seen thus far in every bacterial T4P, they have a completely different packing compared to archaeal filaments (Fig. 4c, d).

A DALI[32] matrix analysis of the C-terminal globular domains (Fig. 5a) shows that archaeal flagellins, T4P and the Iho670 adhesion filament share structural homology for this domain containing an Ig-like β-sandwich. However, this domain in archaea shows no structural similarity to the bacterial T4P, where this globular domain is mainly a β-sheet containing 3-5 β-strands, or almost entirely absent in at least one pilin[33]. A similar matrix was seen when the C-terminal globular domains were analyzed by TM-score[34] (Supplementary Fig. 6). TM-score is a metric for measuring the similarity of protein structures, and has a value in the range (0,1). It is a better metric in this instance, because it is independent of protein length, unlike RMSD analysis. Therefore, archaeal T4P and flagellins cannot be readily distinguished by the fold of the C-terminal domain, but can be easily distinguished simply by the length of the N-terminal helix.

## Discussion

Recent advances in cryo-EM have revolutionized the life sciences[35–37]. Besides yielding near-atomic resolution structures of complex macromolecular assemblies, this technique becomes

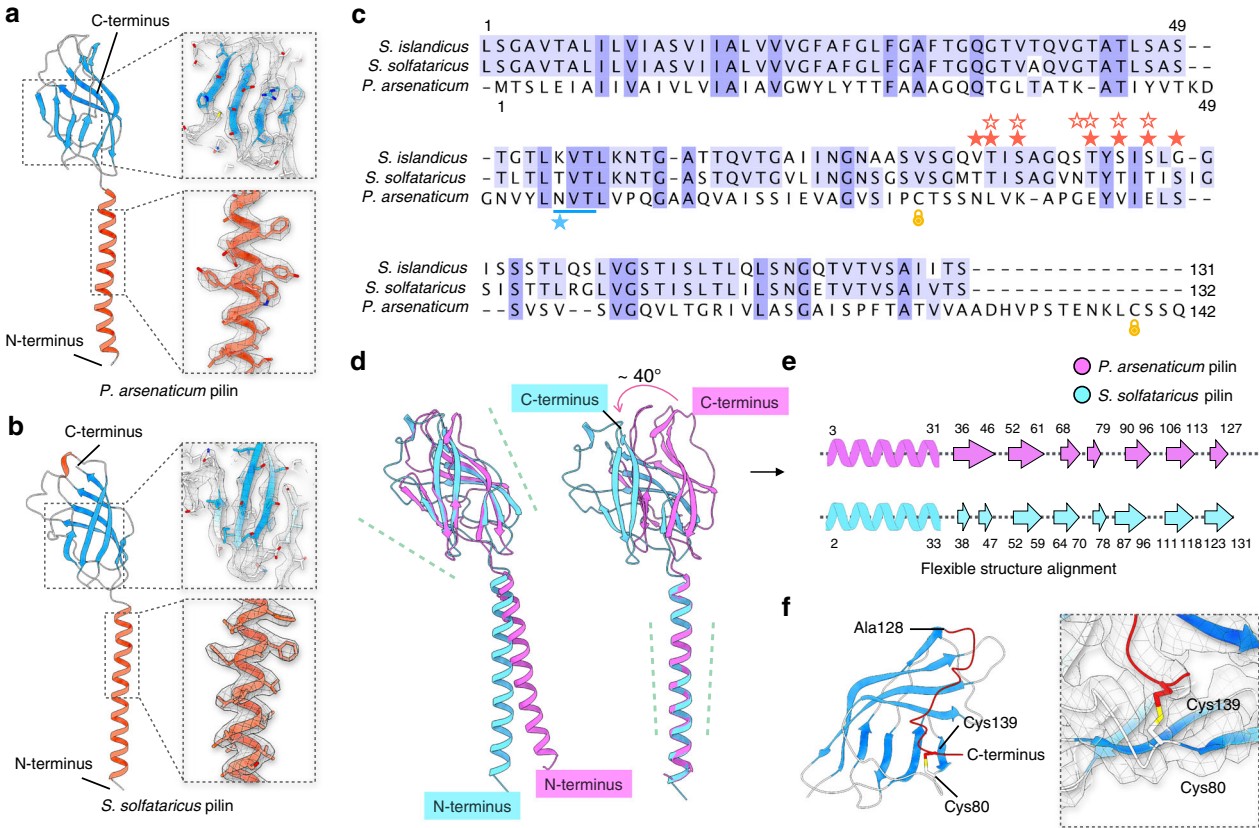

**Fig. 2 Two-domain architecture of archaeal type IV pilus (T4P).** Protein Cα trace of a single *P. arsenaticum* pilin (**a**) and *S. solfataricus* pilin (**b**). Pilin structures are shown in cartoon, colored by secondary structure. The map quality of the N-terminal long helix and C-terminal globular domain are shown on the right. **c** Structure-based sequence alignments of three archaeal type IV pilins with known structures, *P. arsenaticum* pilin, *S. solfataricus* pilin and previously determined *S. islandicus* pilin. Sequences were aligned using Tcoffee[67] and visualized with Jalview[68]. The two blue shades correspond to the amount of sequence identity (dark blue when all three are identical, light blue when two are identical). The observed glycosylated residues are indicated with stars: solid red stars for O-linked sugars in *S. solfataricus*, open red stars for O-linked sugars in *S. islandicus*, and a blue star for N-linked sugar in *P. arsenaticum*. The blue line above the star indicates the NxS/T motif associated with N-linked glycosylation in *P. arsenaticum*. Two cysteine residues that form a disulfide bond in *P. arsenaticum* are indicated with yellow lock symbols. Note that amino acid positions are counted starting with the first amino acid of the mature protein (i.e., following cleavage of the signal peptide). **d** Superposition of a single *P. arsenaticum* pilin (purple) and *S. solfataricus* pilin (cyan), aligned by either the globular domain (left) or the N-terminal helix (right). **e** Flexible secondary structure alignment reveals a simple mapping of the local secondary structure elements between the two pilins. The N-terminal domain is α-helical, and each β-strand in the C-terminal domain is shown as a closed shape. **f** The C-terminal loop of the *P. arsenaticum* pilin and the disulfide bond.

increasingly valuable for de novo identification and characterization of protein components constituting these assemblies, whereby atomic models are built directly into the map densities. This approach has been particularly revealing in the study of complex virions[38–41] and cellular appendages[26]. Here we have used cryo-EM to identify the host receptors recognized by two hyperthermophilic archaeal viruses, SSRV1 and PFV2. Studies on virus-host interactions remain highly challenging in archaea, due to the lack of genetic tools for most organisms, as is the case for the hosts of SSRV1 and PFV2. We have observed that both viruses bind to pili-like filaments and could determine the identity of the respective receptors directly by cryo-EM. We do not know how this would be possible using other approaches. For instance, mass-spectrometry analysis of a crude sample typically yields too many potential candidate proteins for unequivocal identification, whereas some proteins are not amenable to standard tryptic digestion, as has been shown recently for the pili of *S. islandicus*[26].

Pili appear to be a common receptor for hyperthermophilic archaeal viruses and have been previously shown to be recognized by rudiviruses SIRV2 and SIRV8[16,17], and turrivirus STIV[18]. Here we have shown that members of yet another family of archaeal

viruses, the *Tristromaviridae*, also bind to the host cells via T4P. Rudiviruses recognize the adhesive T4P[26], whereas the identity of the pilus receptor of STIV on the surface of *S. solfataricus* has not been determined[18]. We do not exclude the possibility that the *S. solfataricus* T4P identified herein as the receptor for SSRV1 is an ortholog of the pilus recognized by STIV. Notably, direct adsorption experiments with SSRV1 to different members of the genera *Sulfolobus* and *Saccharolobus* showed that the virus could adsorb only to *S. solfataricus* POZ149 and, less efficiently, to *S. solfataricus* P1, but not to *S. islandicus* LAL14/1 or *S. acidocaldarius* DSM 639[27], suggesting that efficient binding is highly specific. Intriguingly, in addition to the divergence of the pilin proteins themselves, there might be species-specific differences in the pilin glycosylation patterns, which could play equally important role in determining the host range of archaeal viruses. Thus, structural characterization of the glycans decorating different archaeal surface structures is likely to shed additional light on virus-host interactions in archaea.

The reasons underlying the selection of T4P as common virus receptors remain unclear. One possible explanation is that T4P are important for survival and functioning of hyperthermophilic archaea in natural habitats and hence the loss of these structures

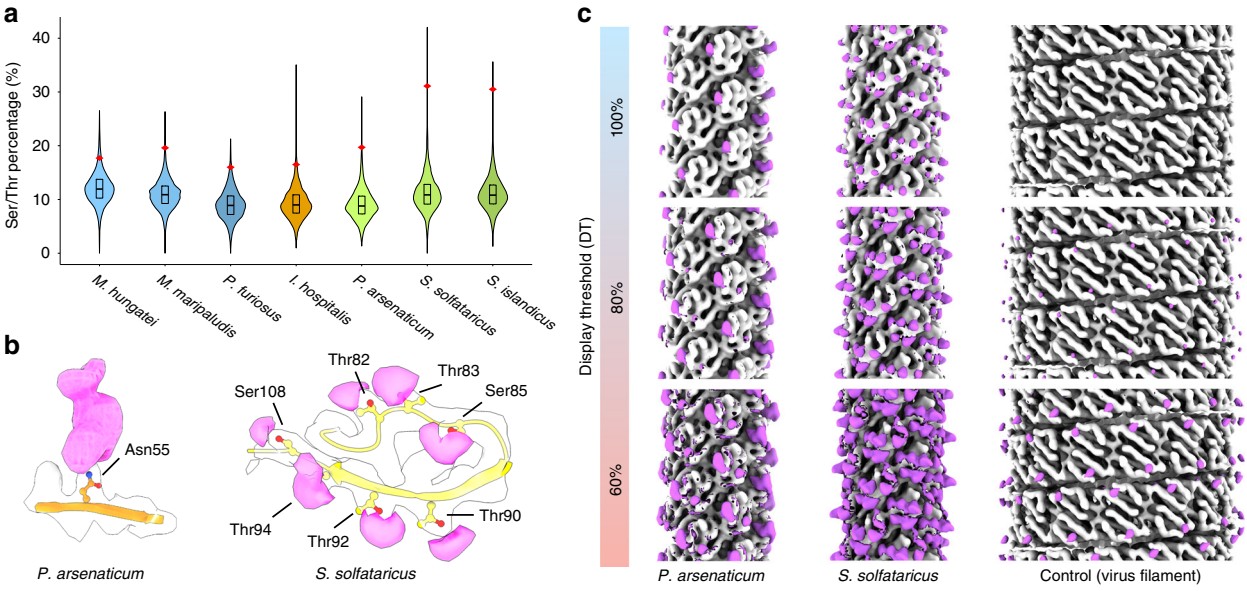

**Fig. 3 Glycosylation of archaeal type IV pilus and other archaeal cell appendages. a** Violin plot showing the distributions of the serine and threonine percentages in all proteins of *M. hungatei, M. maripaludis, P. furiosus, I. hospitalis, P. arsenaticum, S. solfataricus* and *S. islandicus*. The red diamond indicates the percentage of serine+threonine in the pilin or flagellin from this species discussed in this paper. The black box indicates the interquartile range, and the central line in the box indicates the median of the data. The species for the archaeal flagellar filaments are colored in blue-ish colors, adhesion filaments in orange, and archaeal T4P in green-ish colors. **b** Density (magenta) due to post-translational modifications on *P. arsenaticum* pilin (left) and *S. solfataricus* pilin (right). The protein Cα backbone is colored in orange (*P. arsenaticum*) or yellow (*S. solfataricus*). The extra density is due to N-linked (Asn) or O-linked (Thr, Ser) sugars, and the residues modified are labeled. **c** Surface glycosylation of the *P. arsenaticum* pilus (left), *S. solfataricus* pilus (middle) and a control rod-like virus without apparent glycosylation. The control virus and the *S. solfataricus* pilus were purified and imaged together. All three volumes were filtered to 7 Å. The density accounted for by atomic models is colored in gray, and the extra density is colored in magenta. Three different display thresholds are shown. A small additional density appears on the surface of the virus, and this is due to several C-terminal residues that were not part of the atomic model due to disorder, and not due to glycosylation.

would bear a considerable fitness cost. Consistently, T4P are highly widespread in archaea and generally well conserved in members of the same genus[13]. Notably, it has been shown that besides its host *S. solfataricus* POZ149, SSRV1 can deliver its genome, with low efficiency, into a broad range of cells from genera *Saccharolobus/Sulfolobus* and *Acidianus*, even though genome delivery did not lead to virus proliferation[27]. An alternative explanation could be that the inherent properties of T4P might assist, directly or otherwise, in the genome delivery into the cell interior. For instance, T4P might be landmarks for the openings in the S-layer, an external crystalline protein layer surrounding archaeal cells[42,43], which has to be penetrated during the delivery of the viral genome. Furthermore, it has been previously suggested that SIRV2 initially binds to the tips of T4P and progressively moves along the pilus to the cell body, where genome deliver takes place[16]. However, the mechanisms which would promote the virus movement and genome internalization remain to be understood. Additionally, if archaeal T4P retract, as they do in bacteria[1], they would be ideal candidates as receptors for viruses as they could readily lead to the internalization of the viral genome. However, no PilT-like retraction ATPases have been found in archaea, and no direct observations of T4P retraction in archaea have been made thus far[13]. But the Tad pili in *Caulobacter*, which also lack a PilT-like retraction ATPase, have now been shown to retract[44]. We therefore leave this possibility of pilus retraction open.

Being exposed on the cell surface, T4P of hyperthermophilic archaea represent an excellent model for studying the remarkable stability of macromolecular complexes under extreme physico-chemical conditions. Previous study of the *S. islandicus* T4P has revealed that extreme hydrophobicity coupled with extensive glycosylation renders the pilus resistant to various harsh treatments, including boiling in SDS and GuHCl or prolonged treatment with proteases[26]. Comparison of T4P from different hyperthermophiles allows us to dissect further the factors underlying adaptations to different environmental conditions. First, although T4P of a neutrophilic *P. arsenaticum* is as hydrophobic as those of acidophilic *S. islandicus* and *S. solfataricus* (Fig. 2c), it does not possess the extensive sugar coating. This observation suggests that glycosylation plays a more important role for survival in acidic environments. Recently, another adaptation to low pH habitats has been uncovered in archaea-specific glycerol dibiphytanyl glycerol tetraethers (GDGTs) lipids. Headgroups of GDGT in acidophilic hyperthermophiles are covalently linked to calditol, a unique five-membered ring, which is synthesized from a hexose sugar moiety[45]. Decoration of pili and lipids with covalently linked glycans and calditol, respectively, appear to be conceptually similar adaptations to acidic environments. Instead, T4P of *P. arsenaticum* is stabilized by an intramolecular disulfide bond (Fig. 2f), a protein adaptation to high temperatures previously reported in bacteria, archaea and their viruses[46–48]. Disulfides in mesophilic bacterial T4P have also been previously described[49]. Notably, qualitative observations suggest that *P. arsenaticum* T4P do not possess the remarkable resilience observed for *Sulfolobus* T4P (Supplementary Fig. 5), suggesting that it may be glycosylation rather than hydrophobicity which renders the pilus nearly indestructible.

While it was previously suggested that the Iho670 adhesion filaments from the non-motile archaeon *I. hospitalis* were "flagellar-like"[25], we have now shown that all archaeal T4P (or at least, the three structures that now exist) are truly flagellar-like in comparison with the atomic structures for archaeal flagellar filaments. Further, the Ih0670 filaments are much closer to

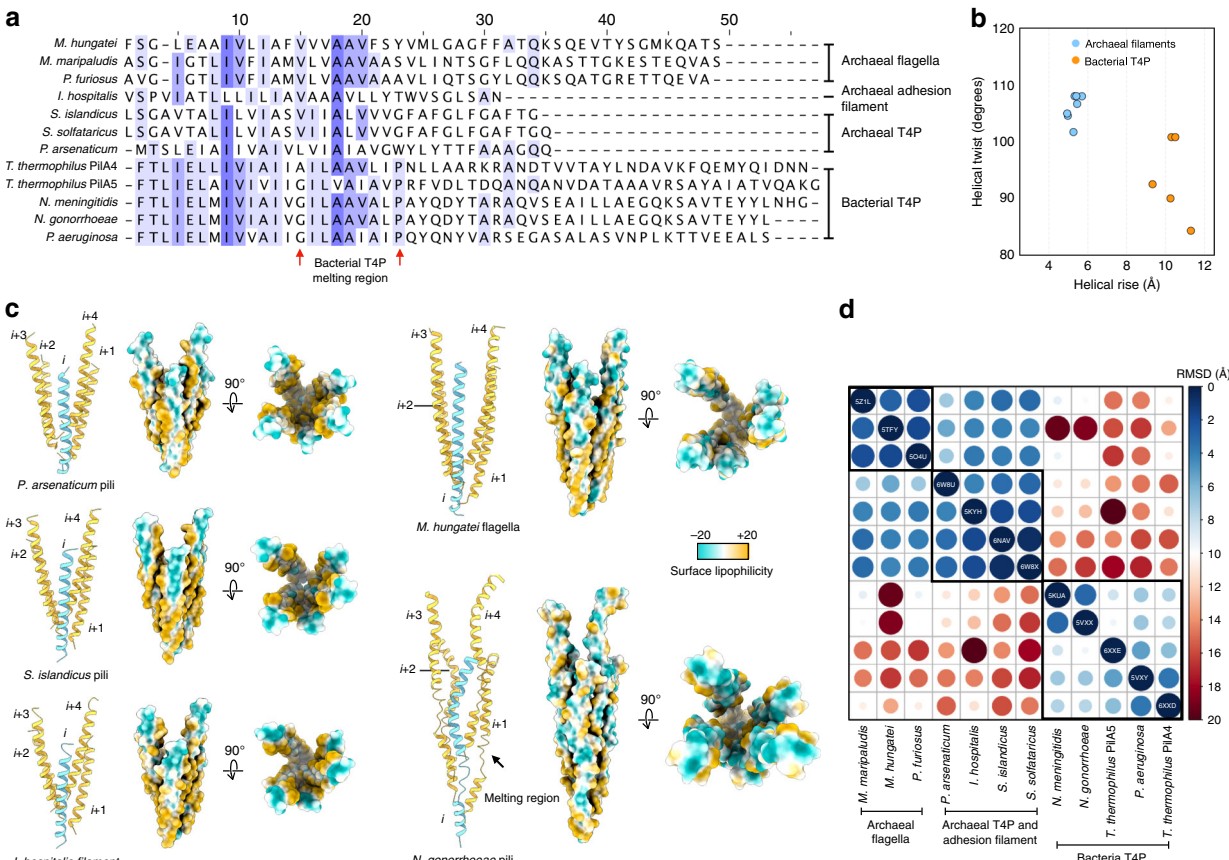

**Fig. 4 Conservation and divergence of the N-terminal helical bundles. a** Sequence alignments of the N-terminal long helix in three archaeal flagellins, one archaeal "adhesion" protein, three archaeal T4P and three bacterial T4P. Area where the bacterial T4P N-terminal helix melts is indicated by two red arrows. **b** Helical symmetries of the filaments mentioned in **a**, with archaeal filaments light blue and bacterial filaments orange. **c** Five adjacent N-terminal helices of *P. arsenaticum* pilins, *S. solfataricus* pilins, *I. hospitalis* proteins, *M. hungatei* flagellins and *N. gonorrhoeae* pilins are shown. The Cα trace is shown in cartoon (left), with one helix colored cyan and the other four colored yellow. The bundles are also shown in an atomic surface view and colored by lipophilicity (right). **d** All-against-all comparison of the five helix bundles of the structures shown in **a**. The matrix is based on the pairwise RMSD comparisons calculated from PyMOL.

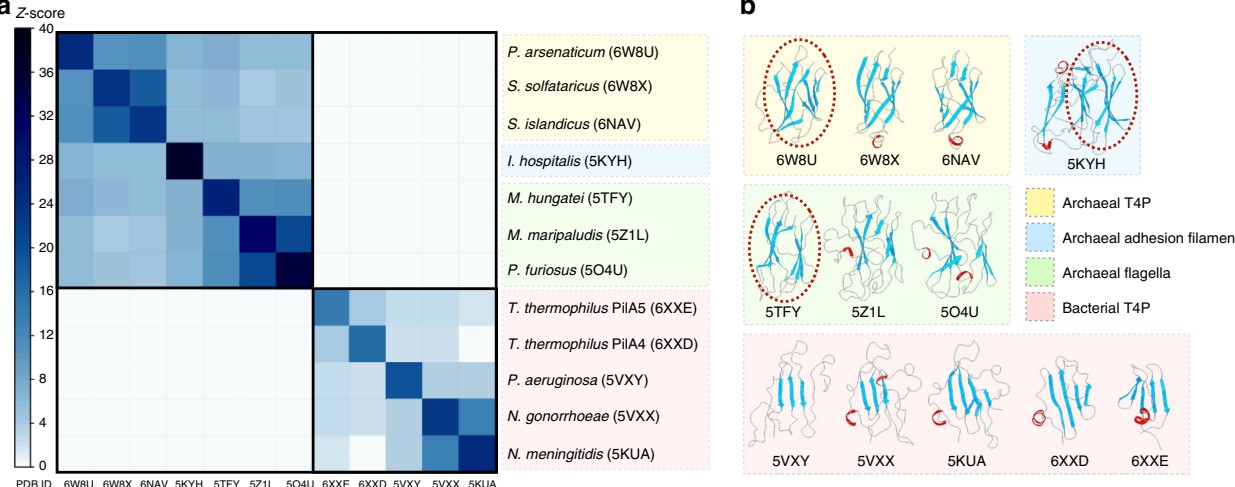

**Fig. 5 Conservation and divergence of the C-terminal globular domains. a** All-against-all comparison of the bacterial T4P and other archaeal structures with N-terminal T4P domains. The N-terminal helices have been removed and only the globular domains were analyzed. The matrix is based on the pairwise Z-score comparisons calculated using the DALI server[32]. The color scale indicates the corresponding Z-scores. Based on their functions and origins, the proteins are partitioned into four groups, which are indicated with different colors on the right. One representative globular domain within each archaeal group was highlighted by a red dashed circle, to show the corresponding homology region. **b** Ribbon diagrams of the globular domains used in **a**.

archaeal T4P than to flagellar filaments, particularly in terms of the N-terminal helix that fails to extend into the C-terminal globular domain. Thus, we think it likely that there may be many other archaeal proteins that are not identified at this point as T4P. Given that the C-terminal domains of T4P-like filaments in bacteria and archaea are unrelated and some bacteria lack this domain altogether[33], it appears likely that the N-terminal domain consisting of the signal sequence and the hydrophobic α-helix, which forms the core of the filament, has evolved first, possibly at the stage of the last universal cellular ancestor (LUCA). In contrast, the C-terminal globular domains have emerged later on, independently in Bacteria and Archaea. Consistent with this, a novel T4P fold has now been found in bacteria, where the C-terminal domain is all α-helical[50]. The archaeal C-terminal domain is clearly related to immunoglobulin-like domains which are widespread in all walks of life[51]. Thus, the contemporary archaeal T4P-like filaments are a result of a fusion of two pre-existing structural domains, a common mechanism of protein evolution and diversification[52,53]. Importantly, a close structural similarity between archaeal pili and flagellar filaments (Fig. 5) strongly suggests that functional diversification of the two types of archaeal filaments took place following the split of bacterial and archaeal lineages from their common ancestor.

## Methods

**Strain cultivation and filament purification.** Exponentially growing cultures of *Saccharolobus solfataricus* POZ149[27] and *Pyrobaculum arsenaticum* 2GA[54] cells were infected with fresh preparations of Saccharolobus solfataricus rod-shaped virus 1 (SSRV1) and Pyrobaculum filamentous virus 2 (PFV2), respectively. The infected culture of *S. solfataricus* POZ149 was incubated at 75 °C under agitation for 2 days, whereas *P. arsenaticum* 2GA culture was grown during 4 days at 90 °C without shaking. After the removal of cells (7438 × g, 20 min, Sorvall 1500 rotor), viruses along with pili detached from the cells were collected and concentrated by ultracentrifugation (273,620 × g, 2 h, 10 °C, Beckman 126 SW41 rotor). For cryo-EM analysis, the concentrated particles were resuspended in buffer A[55]: 20 mM KH2PO4, 250 mM NaCl, 2.14 mM MgCl2, 0.43 mM Ca(NO3)2 and <0.001% trace elements of Sulfolobales medium, pH 6. Virus particles and pili also copurified when the pelleted virion-pili mixtures were subjected to centrifugation in a CsCl buoyant density gradient (0.45 g ml−1) with a Beckman SW41 rotor at 260,110 x g for 20 h at 10 °C.

**DNA extraction, sequencing and genome analysis.** DNA was extracted from exponentially growing cultures of *S. solfataricus* POZ149 and *P. arsenaticum* 2GA using the Wizard Genomic DNA Purification Kit (Promega). The genome sequence of *S. solfataricus* POZ149 was obtained using PacBio single-molecule real-time (SMRT) technology, while libraries with 250 bp insert size were sequenced on Illumina platform for *P. arsenaticum* 2GA (Novogen, China). Raw sequence reads were processed and assembled with FALCON[56] and SOAPdenovo version 2.04[57], respectively. Gene prediction was performed by GeneMark.hmm version 3.25[58]. The genome sequence of *S. solfataricus* POZ149 was deposited in GenBank under the accession number CP050869. The Whole Genome Shotgun project for *P. arsenaticum* 2GA has been deposited at DDBJ/ENA/GenBank under the accession JAAVJF000 000000. The version described in this paper is version JAAVJF010000000.

**Negative-stain electron microscopy.** For TEM, 5 µl of the samples were applied to carbon-coated copper grids and negatively stained with 2% uranyl acetate. The stained grids were imaged with the transmission electron microscope FEI Tecnai Biotwin (Institut Pasteur, France).

**Cryo-electron microscopy and image analysis.** The pilus sample (~4.0 µl) was applied to discharged lacey carbon grids and plunge frozen using a Vitrobot Mark IV (FEI). Frozen grids were imaged in a Titan Krios at 300 keV and recorded with a K3 camera at 1.08 Å/pixel. The *P. arsenaticum* pilus data were collected in the UVA core facility, and the *S. solfataricus* pilus was collected in National Cryo-EM Facility (NCEF) of the National Cancer Institute (NCI). Micrographs were collected using a defocus range of 1–2 µm, with a total exposure dose of ~50 electrons/Å² distributed into ~25 fractions. To get a preliminary helical reconstruction volume in SPIDER[59], all of the micrographs were first motion corrected and dose weighted by MotionCorr v2, and then CTF-multiplicated by the theoretical CTF. Filament images amounting to ~20 electrons /Å² were extracted by EMAN2[60]. A small subset containing ~50,000 overlapping 384-pixel-long segments was used to search for the correct helical symmetry. For *P. arsenaticum* we explored a multiplicity of

symmetries that were each consistent with the averaged power spectrum. The number of possible symmetries was six (Supplementary Table 2), and each of these was tested in SPIDER[59] using IHRSR[61], until recognizable protein features were seen, such as side-chains. A ~4 Å reconstruction was generated from this small subset with the correct symmetry (entry #1, Supplementary Table 2), and this volume was subsequently filtered to 7 Å as the starting reference used in RELION[62]. For *S. solfataricus*, the high degree of sequence identity with the *S. islandicus* LAL14/1 pilus suggested that a helical symmetry close to that previously determined[26] for *S. islandicus* might be the correct one, and this turned out to be the case. After using the full dataset in RELION (210,331 segments for the *P. arsenaticum* pilus, 579,713 segments for the *S. solfataricus* pilus, Table 1), doing CTF-refinement and Bayesian polishing, the final volumes were estimated to have a resolution of 3.8 Å for the *P. arsenaticum* pilus and 3.4 Å for the *S. solfataricus* pilus, based on the map:map FSC, model:map FSC and d99[30]. The final volumes were then sharpened with a negative B-factor automatically estimated in RELION (Table 1).

**Model building.** The density corresponding to a single *S. solfataricus* pilin was segmented from the experimental cryo-EM density map using Chimera[63]. Then a *S. solfataricus* pilin model was generated by homology modeling using the *S. islandicus* LAL14/1 pilin as the reference, and then docked into the segmented map. Then this model was adjusted manually in Coot and real-space refined in PHENIX[64]. Finally, the refined single pilin model was used to generate a filamentous model using the determined helical symmetry, and this filament model was refined against the full cryo-EM map using in PHENIX. MolProbity[65] was used to evaluate the quality of the filament model. The refinement statistics are shown in Table 1.

The density corresponding to a single *P. arsenaticum* pilin was also segmented in the same manner. Due to the uncertainty in the pilin sequence, seven potential pilin candidates were selected based on secondary structure predictions (WP_011899639, WP_011900226, WP_011899654, WP_011900935, WP_011900944, WP_011900938, and WP_011900942). Each of the sequences was used to do de novo model building against the segmented map using Rosetta CM[66]. As a result, only HC235_07175 could be built into the cryo-EM map and was also detected by MS/MS analysis. To further validate the sequence of both pilin, the whole genomes of both strains were sequenced and deposited. The model building for the assembled *P. arsenaticum* pilus was the same as described above.

**Mass spectrometry.** The solution samples containing the pili were reduced with 10 mM DTT in 0.1 M ammonium bicarbonate then alkylated with 50 mM iodoacetamide in 0.1 M ammonium bicarbonate (both room temperature for 0.5 h). The sample was digested overnight at 37 °C with 0.1 µg (Chymotrypsin, Asp-N and Trypsin) in 50 mM ammonium bicarbonate. The sample was acidified with acetic acid to stop digestion and then spun down. The solution was evaporated to 15 µL for MS analysis.

The LC-MS system consisted of a Thermo Electron Q Exactive HF-X mass spectrometer system with an Easy Spray ion source connected to a Thermo 75 µm x 15 cm C18 Easy Spray column. 7 µL of the extract was injected and the peptides eluted from the column by an acetonitrile/0.1 M formic acid gradient at a flow rate of 0.3 µL/min over 1.0 hours. The nanospray ion source was operated at 1.9 kV. The digest was analyzed using the rapid switching capability of the instrument acquiring a full scan mass spectrum to determine peptide molecular weights followed by product ion spectra (10 HCD) to determine amino acid sequence in sequential scans. This mode of analysis produces approximately 25,000 MS/MS spectra of ions ranging in abundance over several orders of magnitude. Not all MS/MS spectra are derived from peptides. The data were analyzed by database searching using the Sequest search algorithm against Uniprot *Sulfolobus solfataricus* or Uniprot *Pyrobaculum arsenaticum*.

## Data availability

The genome sequence of *S. solfataricus* POZ149 was deposited in GenBank under the accession number CP050869. The Whole Genome Shotgun project for *P. arsenaticum* 2GA has been deposited at DDBJ/ENA/GenBank under the accession number JAAVJF000000000. The version described in this paper is JAAVJF010000000. The atomic model for the *P. arsenaticum* pilus was deposited at the Protein Data Bank with accession code 6W8U, and the corresponding map was deposited at the Electron Microscopy Data Bank with code EMD-21578. The atomic model for the *S. solfataricus* pilus was deposited at the Protein Data Bank with accession code 6W8X, and the corresponding map was deposited at the Electron Microscopy Data Bank with code EMD-21579. Other data are available from the corresponding authors upon reasonable request.

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

## Acknowledgements

This research was, in part, aided by the National Cancer Institute's National Cryo-EM Facility at the Frederick National Laboratory for Cancer Research under contract HSSN261200800001E. Cryo-EM imaging was also conducted at the Molecular Electron Microscopy Core facility at the University of Virginia, which is supported by the School of Medicine and built with NIH grant G20-RR31199. This work was supported by NIH grant R35GM122510 (to E.H.E.). M.K. was supported by l'Agence Nationale de la Recherche (France) project ENVIRA. D.P.B. is part of the Pasteur – Paris University (PPU) International PhD Program, which has received funding from the European Union's Horizon 2020 research and innovation programme under the Marie Sklodowska-Curie grant agreement No 665807. We are also grateful to the Ultrastructural BioImaging (UTechS UBI) unit of Institut Pasteur for access to electron microscopes.

## Author contributions

D.P.B. prepared all of the samples; F.W. collected the cryo-EM data; F.W., L.B. and E.H.E. performed image processing; F.W. did the structural modeling; F.W., Z.S. and M.K. prepared figures; F.W., M.K. and E.H.E. wrote the manuscript; E.H.E., D.P. and M.K. conceived the study.

## Competing interests

The authors declare no competing interests.
