## [Peer Review File · Nature Communications]

Reviewer #1 (Remarks to the Author):

Wang et al present two new archaeal type 3 pili (T4P) structures determined by cryo-EM, which add to the single presently known archaeal T4P structure and enables them to compare all known structures of bacterial and archaeal T4P structure with each other and archaeal flagellar filaments. They identify glycosylation as a key factor for the high thermal stability of T4P and describe their interaction with helical viruses. Without leaning on prior structural information, they remarkably succeeded by sequencing the genomes of these archaea and by combining mass spec and bioinformatics to build de-novo models into the cryo-EM maps. Combining these multi-disciplinary analyses, the authors propose an intriguing hypothesis about a common evolutionary origin between T4P and archaeal filaments.

This is an excellent paper, well-written and packed with clearly arranged dense information. I have thoroughly enjoyed reading it. The figures are excellent. The presentation is logical and easy to follow. It is one of the rare cases I recommend publishing without any significant further modifications.

Minor comments:

Line 439: "b", use bold face

Figs. 4d and 5a: I really like these. Well done!

Line 308: "The helical symmetry was determined in SPIDER using IHRSR after searching through all possible symmetries by trial and error". This statement should be expanded to explain the search strategy more comprehensively. There are many local maxima in helical symmetry searches and it would be computationally expensive to explore them all. At the minimum, it should be stated that a local search around the helical symmetry shared by the known archaeal filament structures yielded a solution that refined to high resolution.

Lines 300ff: this paragraph apparently contains some unprintable characters (e.g. 300?keV)

Lines 256-258: I don't follow the argument why the NTD would have evolved first. Please clarify.

Line 248: "suggesting that it is glycosylation" -> tone down a notch to "suggesting that it may be glycosylation"

Signed: M.Wolf

Reviewer #2 (Remarks to the Author):

This well written paper by Wang et al. presents structures of two archaeal T4Ps from *P. arsenaticum*, a neutrophilic hyperthermophile and *S. solfataricus*, an acidophilic hyperthermophile, after co-purification with their respective viral partners and characterization by cryo-electron microscopy (cryo-EM) to identify the relevant gene product among multiple possibilities as they have done previously. The authors compare these structures to other filaments from the same superfamily to understand some of the distinguishing characteristics. Comparisons of the amount of glycosylation were used to explain differences in stability of these filaments in harsh conditions, particularly low pH. This work adds two more structures of archaeal T4Ps for structural comparisons to explain evolutionary lineages, including more confident assignment of Iho670 adhesion filaments as T4P. Despite the use of phage to identify the relevant filaments, the potential role that glycosylation plays in affecting the viral-T4P interaction was not addressed. Do the phages bind to the archaeal pilins directly or to the sugars? The three themes of viral-host

interaction, glycosylation of the two T4Ps and structural comparisons of the archaeal T4Ps with bacterial T4Ps and archaeal flagella could be better connected.

Comments:

Line 59, ectodomain is an odd term to use

Line 103, too many significant figures. The opposite 7-start nomenclature has the potential to be confusing to readers not familiar with this filament family, especially since both are right handed one start helices.

Line 152, are pilin glycosylation enzymes encoded with the pilin genes?

In the third paragraph of results, the authors mention that the C-terminal domains are structurally conserved, except "one β -strand in *P. arsenaticum* maps to two shorter β -strands in *S. solfataricus*". What is the functional significance of this difference, is it conserved in each group of archaea?

Line 188, electrostatic interactions

Line 230, Tad pili extend and retract with only a single ATPase (shown by the Brun lab for *Caulobacter*), this is not unprecedented

Line 246, disulfide bonds are common in bacterial type IV pilins, both at the C-terminus and internal in some species – this should be mentioned, as written it sounds more unique than it is

Line 255, I don't understand this statement based on the preceding evidence

Figure 2c descriptions: specify the sequence alignment program used and what the different shades of blue correspond to in the alignment map

Figure 2e: use arrows to represent β -sheets instead of circles

Figure 2a/b/f: indicate the N and C-termini

Many comparisons were made between *S. solfataricus*/*S. islandicus* and *P. arsenaticum* pilins including % of charged residues, O-/N-linked glycosylation sites, sequence identities, etc. To help readers appreciate these differences, a table or schematic showing side-by-side comparisons of these parameters would help

Figure 3 legend descriptions of panels a and b are switched

Figure 3a: what do blue, orange, green represent?

Supplementary figure legends 3 and 4 are switched

In the seventh paragraph of results, figure identification should be (Fig. 4a,c) instead of (Fig. 3a,c)

In the third paragraph of discussion, the supplementary figure was incorrectly labelled. Should be (Supp. Fig. 4) not (Supp. Fig. 3)

Is it necessary to include subheadings for each of the results?

Reviewer #3 (Remarks to the Author):

In their manuscript Wang et al. report two new cryo-EM structures of archaeal Type IV pili, that of *Sulfolobus islandicus* and *Pyrobaculum arsenaticum*, and demonstrate these structures represent the receptors for two filamentous phages recently identified by the authors. The new T4P structures allow a robust analysis of the phylogenetic relationship and structure-function aspects of bacterial and archaeal T4P as well as the T4P-related archaeal flagellum, also known as "archaellum". In addition, the authors discuss the role of T4P glycosylation in function of the chemical and high temperature resistance of the fibers.

The paper is well written and brings new insights in the evolutionary relationships of T4P-like filaments that should be of interest to a diverse audience.

Overall, the paper is technically sound and the data are supportive of the conclusions presented, with the exception of the proposed claims regarding the high stability of the fibers, which would require some additional experiment to be conclusive. This reflects the single major comments I have, which is the lack of systematic experimental interrogation of the structural aspects that provide the fibers' strength. There is no direct demonstration that the glycosylation or the C-terminal extension in T4P of *S. solfataricus* and *P. arsenaticum* provides the fibers' resistance to acidic or high temperature conditions, respectively. The only data shown is qualitative in nature (Suppl. Fig. 4) and shows single negative stain EM images only. A more quantitative and comprehensive analysis of the chemical and heat stability of the fibers would be advisable. For example by monitoring the release – or lack thereof – of subunits by SDS-PAGE.

Specific comments and suggestions:

1. In Ln 90-100 and Figure 1 the authors convincingly demonstrate that T4P pili represent the receptor for filamentous phages SSRV1 and PFV2. Presumably these interactions are specific for the cognate T4P – phage pair? It would be good to evaluate this and mention it in the paper. In the discussion the authors propose that the targeting of T4P by the phages may be related to the strong conservation of the pili or a possibility for the phages to move along the length of the fiber to gain access to the cell. Another aspect worth considering is that T4P may represent a location corresponding to a breach in the S-layer, which may facilitate access to the cytoplasmic membrane.
2. In lines 118-128 the authors derive and discuss the cleavage sites of the signal peptides in prepilins. It would be worth highlighting this finding in an additional figure panel. For example as part of Suppl. Figure 2.
3. Lines 131-136 and in Fig 2e, the authors compare structural similarity in the T4P N- and C-terminal domains. It would be helpful to provide RMSD values for the alignment of the respective domains to give some quantitative measure of the similarity.
4. In Ln 139-142 and Fig. 2f the authors describe the presence of a disulfide-bonded C-terminal extension in *P. arsenaticum*. It would be helpful to also illustrate the location of the disulfide linked C-terminal extension in the context of the full fiber. Is there a possibility to form intermolecular crosslinks amongst subunits, or does the structure only allow for intramolecular crosslinking?
5. In Ln 144-147 *S. solfataricus* and *S. islandicus* pili are said to have 37% Ser/Thr in the globular domains. This does not seem to correspond with the data shown in Fig. 3a, which appears to show more 30-31% in Ser/Gly. Please correct or clarify the difference.
6. In Ln 149-160 the authors describe the indications of glycosylation in the two T4P structures solved. It would be good to mention whether glycosylation is also seen in the MS fingerprint peptides, and if not, why.
7. Unless for my oversight, I did not find Suppl. Figure 3 cited in the main text, and in line 165, "Suppl. Fig. 3" should be "Suppl. Fig. 4). Also note that in Figure 3, the caption of panels a and b does not correspond with the order of the panels in the Figure.
8. In Jarrel and Albers 2012 (ref <https://doi.org/10.1016/j.tim.2012.04.007>) the name "archaellum" was proposed for the "archaeal flagellar filaments". This reviewer is of the opinion that that proposed nomenclature does well in highlighting the fundamental difference in bacterial and archaeal flagella, and would advise the authors to adopt the nomenclature in their manuscript.

We are very thankful to each of the three reviewers for the extremely careful reading of our paper. Only one substantial issue was raised (by Reviewer #3) and we think that we have addressed this and the more minor issues in this Response and the revised paper.

Reviewer #1

Minor comments:

Line 439: “b”, use bold face

Done

Figs. 4d and 5a: I really like these. Well done!

Thank you.

Line 308: “The helical symmetry was determined in SPIDER using IHRSR after searching through all possible symmetries by trial and error”. This statement should be expanded to explain the search strategy more comprehensively. There are many local maxima in helical symmetry searches and it would be computationally expensive to explore them all. At the minimum, it should be stated that a local search around the helical symmetry shared by the known archaeal filament structures yielded a solution that refined to high resolution.

This is a very good suggestion to explain better what was done. For *P. arsenaticum* we did not use any knowledge about known archaeal filament structures and actually explored a multiplicity of symmetries that were each consistent with the averaged power spectrum. The number of possible symmetries was six (Supp. Table 2), and each of these was tested in SPIDER using IHRSR, until recognizable protein features were seen, such as protein side-chains. Thus, such a search is computationally expensive, but easily achievable with current computers. The computing time for searching each symmetry is about three days using a workstation with 30 CPUs, and these searches were run in parallel on a cluster. Thus, the extensive search only took ~ three days in total. For *S. solfataricus*, the high degree of sequence identity with the *S. islandicus* LAL14/1 pilus suggested that a helical symmetry close to that previously determined (Wang et al., 2019) for *S. islandicus* might be the correct one, and this turned out to be the case. This has now been added into Methods.

Lines 300ff: this paragraph apparently contains some unprintable characters (e.g. 300?keV)
Fixed.

Lines 256-258: I don't follow the argument why the NTD would have evolved first. Please clarify.

We have clarified the statement, as follows:

“Given that the C-terminal domains of T4P-like filaments in bacteria and archaea are unrelated and some bacteria lack this domain altogether (Reardon and Mueller, 2013), it appears likely that the N-terminal domain consisting of the signal sequence and the hydrophobic α -helix,

which forms the core of the filament, has evolved first, possibly at the stage of the last universal cellular ancestor (LUCA).”

Line 248: “suggesting that it is glycosylation” -> tone down a notch to “suggesting that it may be glycosylation”

Done

Reviewer #2 (Remarks to the Author):

This well written paper by Wang et al. presents structures of two archaeal T4Ps from *P. arsenaticum*, a neutrophilic hyperthermophile and *S. solfataricus*, an acidophilic hyperthermophile, after co-purification with their respective viral partners and characterization by cryo-electron microscopy (cryo-EM) to identify the relevant gene product among multiple possibilities as they have done previously. The authors compare these structures to other filaments from the same superfamily to understand some of the distinguishing characteristics. Comparisons of the amount of glycosylation were used to explain differences in stability of these filaments in harsh conditions, particularly low pH. This work adds two more structures of archaeal T4Ps for structural comparisons to explain evolutionary lineages, including more confident assignment of Iho670 adhesion filaments as T4P. Despite the use of phage to identify the relevant filaments, the potential role that glycosylation plays in affecting the viral-T4P interaction was not addressed. Do the phages bind to the archaeal pilins directly or to the sugars?

Unfortunately, we have no way to address this, as pili cannot be produced that are not glycosylated. Our previous attempts to de-glycosylate the pili enzymatically were unsuccessful, whereas chemical deglycosylation results in aggregation and insoluble filaments, presumably due to their high hydrophobicity. These results are reported in our previous publication (Wang et al., 2019).

The three themes of viral-host interaction, glycosylation of the two T4Ps and structural comparisons of the archaeal T4Ps with bacterial T4Ps and archaeal flagella could be better connected.

Thank you for this suggestion. In the revised manuscript, we made an effort to improve the flow between the three themes, in particular, between virus-host interaction and glycosylation. The glycosylation part is now preceded by the following passage:

“Despite high sequence similarity between the pilins of *S. islandicus* and *S. solfataricus*, SSRV1 does not bind to *S. islandicus* pili and, accordingly, cannot infect this organism²⁷. Given that the surface of the *S. islandicus* pilus is coated with a glycan layer²⁶, the latter might play an important, if not key, role in virus binding. Thus, we next analyzed the glycosylation status of the *S. solfataricus* and *P. arsenaticum* pili.”

Comments:

Line 59, ectodomain is an odd term to use

We have now used “C-terminal globular domain”. It is only an ectodomain when the N-terminal domain is embedded in a membrane.

Line 103, too many significant figures.

We have reduced the number of significant figures in the text but kept them in the table. The reason for this is that this level of precision is needed to properly fit many subunits into the volume.

The opposite 7-start nomenclature has the potential to be confusing to readers not familiar with this filament family, especially since both are right handed one start helices.

While we agree that this might be potentially confusing, one of the most striking differences in the surfaces of these two filaments is the opposite hand of these 7-start helices.

Line 152, are pilin glycosylation enzymes encoded with the pilin genes?

The specific glycosyltransferase implicated in the glycosylation of pili in *Sulfolobus/Saccharolobus* remains unknown. More generally, to the best of our knowledge, enzymes responsible for O-glycosylation, as observed in the case of *Sulfolobus* and *Saccharolobus* pili, have not been identified in any archaea. There are no obvious candidates in the vicinity of the pilin genes, either.

In the third paragraph of results, the authors mention that the C-terminal domains are structurally conserved, except “one β -strand in *P. arsenaticum* maps to two shorter β -strands in *S. solfataricus*”. What is the functional significance of this difference, is it conserved in each group of archaea?

It is hard to see any potential functional significance. We cannot say whether this is conserved in each group until atomic structures are produced for many more members of each group.

Line 188, electrostatic interactions

We think that the present mention of the charged Glu in bacterial T4P is sufficient, as we go on to say: “Because of the melted N-terminal region seen thus far in every bacterial T4P, they have a completely different packing compared to archaeal filaments (Fig. 4c-d).” Thus, there seems little point in discussing the electrostatics of the bacterial T4P.

Line 230, Tad pili extend and retract with only a single ATPase (shown by the Brun lab for *Caulobacter*), this is not unprecedented

We have added a reference to the Tad pili.

Line 246, disulfide bonds are common in bacterial type IV pilins, both at the C-terminus and internal in some species – this should be mentioned, as written it sounds more unique than it is

We now mention this and have added a reference.

Line 255, I don't understand this statement based on the preceding evidence

The Iho670 "adhesion filaments" were not recognized initially as T4P. We expect that there are other such filament-forming proteins that assemble into T4P that are not currently recognized by sequence searches.

Figure 2c descriptions: specify the sequence alignment program used and what the different shades of blue correspond to in the alignment map

Sequences were aligned using Tcoffee and visualized with Jalview. The two blue shades correspond to amount of sequence identity (dark blue when all three are identical, light blue when two are identical). We have added this to the figure legend.

Figure 2e: use arrows to represent β -sheets instead of circles

Done.

Figure 2a/b/f: indicate the N and C-termini

Done.

Many comparisons were made between *S. solfataricus*/*S. islandicus* and *P. arsenaticum* pilins including % of charged residues, O-/N-linked glycosylation sites, sequence identities, etc. To help readers appreciate these differences, a table or schematic showing side-by-side comparisons of these parameters would help

We have provided this information in Supplementary Table 1.

Figure 3 legend descriptions of panels a and b are switched

Fixed

Figure 3a: what do blue, orange, green represent?

Each violin plot corresponds to the serine+threonine distribution in one archaeal species. The archaeal species that produce archaeal flagellar filaments are colored in blue-ish colors, adhesion filaments in orange, and archaeal T4P in green-ish colors. We have added this to the figure legend.

Supplementary figure legends 3 and 4 are switched

It was actually the references to these figures in the text that were swapped. This has been fixed.

In the seventh paragraph of results, figure identification should be (Fig. 4a,c) instead of (Fig. 3a,c)

Fixed

In the third paragraph of discussion, the supplementary figure was incorrectly labelled. Should

be (Supp. Fig. 4) not (Supp. Fig. 3)

Fixed

Is it necessary to include subheadings for each of the results?

We did not use subheadings and thought that they were not necessary. However, we will follow the editor's recommendations.

Reviewer #3 (Remarks to the Author):

In their manuscript Wang et al. report two new cryo-EM structures of archaeal Type IV pili, that of *Sulfolobus islandicus* and *Pyrobaculum arsenaticum*, and demonstrate these structures represent the receptors for two filamentous phages recently identified by the authors. The new T4P structures allow a robust analysis of the phylogenetic relationship and structure-function aspects of bacterial and archaeal T4P as well as the T4P-related archaeal flagellum, also known as "archaellum". In addition, the authors discuss the role of T4P glycosylation in function of the chemical and high temperature resistance of the fibers.

The paper is well written and brings new insights in the evolutionary relationships of T4P-like filaments that should be of interest to a diverse audience.

Overall, the paper is technically sound and the data are supportive of the conclusions presented, with the exception of the proposed claims regarding the high stability of the fibers, which would require some additional experiment to be conclusive. This reflects the single major comments I have, which is the lack of systematic experimental interrogation of the structural aspects that provide the fibers' strength. There is no direct demonstration that the glycosylation or the C-terminal extension in T4P of *S. solfataricus* and *P. arsenaticum* provides the fibers' resistance to acidic or high temperature conditions, respectively. The only data shown is qualitative in nature (Suppl. Fig. 4) and shows single negative stain EM images only. A more quantitative and comprehensive analysis of the chemical and heat stability of the fibers would be advisable. For example by monitoring the release – or lack thereof – of subunits by SDS-PAGE.

We never intended to show in this paper the role of glycosylation in providing the remarkable stability of these filaments. This was done for the *Sulfolobus islandicus* LAL14/1 filaments in our previous recent paper, "An extensively glycosylated archaeal pilus survives extreme conditions", (Wang et al., 2019). The problem is that once the glycosylation is chemically removed (all attempts to remove the glycans enzymatically have failed), the filaments are insoluble (and cannot be run on gels) as would be predicted by their extremely hydrophobic protein surface. What we did show in the present manuscript is that under the conditions used in our previous paper the *Pyrobaculum arsenaticum* filaments are not stable, but the *Sulfolobus islandicus* LAL14/1 filaments were stable. The reviewer suggests SDS-PAGE analysis to monitor the stability of the filaments. As we showed in the previous paper, we were only able to see trace amounts of the pilin by SDS-PAGE using silver staining. By conventional Coomassie blue staining, no bands were ever visible under any conditions. One cannot do any quantitative

analysis of silver-stained gels. Thus, no further experiments are really possible. Given that pili of *S. solfataricus* and *S. islandicus* are 81% identical, we considered that repeating the same set of experiments on the stability of *S. solfataricus* pilus was not warranted because they are not expected to provide any new information compared with the previously published results. Furthermore, *Pyrobaculum arsenaticum* 2GA grows very poorly under laboratory conditions compared to *S. solfataricus* (OD600 never reaches more than 0.2-0.25). Thus, it is highly challenging to obtain sufficient amount of pure pili material for comparative biochemical studies. Finally, due to the pandemic, our laboratories are starting to reopen only now, with the activities of all technical platforms (e.g., proteomics) being reserved for COVID-19 projects. Thus, performing any extensive additional experimentation remains highly problematic. However, we do agree with the reviewer that more quantitative analysis would be desirable. In the revised version, we softened the claims about this aspect throughout the manuscript, including in the Discussion and explicitly state that the observations are qualitative (e.g., “qualitative observations suggest that *P. arsenaticum* T4P do not possess the remarkable resilience”).

Specific comments and suggestions:

1. In Ln 90-100 and Figure 1 the authors convincingly demonstrate that T4P pili represent the receptor for filamentous phages SSRV1 and PFV2. Presumably these interactions are specific for the cognate T4P – phage pair? It would be good to evaluate this and mention it in the paper. Indeed, host recognition is highly specific. We have previously performed adsorption experiments with SSRV1 to different members of the genera *Sulfolobus/Saccharolobus* (Baquero, 2020) and the virus could adsorb only to *S. solfataricus* POZ149 and, less efficiently, to *S. solfataricus* P1, but not to *S. islandicus* LAL14/1 or *S. acidocaldarius* DSM 639. This is now mentioned in the Discussion.

In the discussion the authors propose that the targeting of T4P by the phages may be related to the strong conservation of the pili or a possibility for the phages to move along the length of the fiber to gain access to the cell. Another aspect worth considering is that T4P may represent a location corresponding to a breach in the S-layer, which may facilitate access to the cytoplasmic membrane.

Thank you for pointing out this interesting possibility. It is now mentioned in the Discussion, as suggested:

“T4P might serve as landmarks for openings in the S-layer, an external crystalline protein layer surrounding archaeal cells (Kandiba and Eichler, 2014; Klingl et al., 2019), which has to be penetrated during the delivery of the viral genome.”

2. In lines 118-128 the authors derive and discuss the cleavage sites of the signal peptides in prepilins. It would be worth highlighting this finding in an additional figure panel. For example as part of Suppl. Figure 2.

Thank you for the suggestion. We have added two panels to Supplementary Figure 2 showing the aligned cleavage sites for representative *Pyrobaculum* and *Sulfolobales* species.

3. Lines 131-136 and in Fig 2e, the authors compare structural similarity in the T4P N- and C-terminal domains. It would be helpful to provide RMSD values for the alignment of the respective domains to give some quantitative measure of the similarity.

We have added TM-score analysis into Supplemental Figure 6. RMSD analysis is very sensitive to how many atoms were aligned, while the TM-score introduces a scale to normalize the distance errors and makes the value of the TM-score length-independent for random structure pairs. TM-score has a value in the range (0,1), where 1 indicates a perfect match between two structures. Based upon statistics of structures in the PDB, scores below 0.17 correspond to randomly chosen unrelated proteins whereas structures with a score higher than 0.5 indicate generally the same fold.

4. In Ln 139-142 and Fig. 2f the authors describe the presence of a disulfide-bonded C-terminal extension in *P. arsenaticum*. It would be helpful to also illustrate the location of the disulfide linked C-terminal extension in the context of the full fiber. Is there a possibility to form intermolecular crosslinks amongst subunits, or does the structure only allow for intramolecular crosslinking?

The pilin is already properly folded and stored in the membrane before it gets assembled into pili by the secretion system. So it is hard to imagine how intermolecular crosslinks could exist in the pili unless there was a disulfide bond formation enzyme within the secretion system. Nevertheless, we put a figure in the Supplemental to show the cysteine locations in the filament (Supp. Fig. 4).

5. In Ln 144-147 *S. solfataricus* and *S. islandicus* pili are said to have 37% Ser/Thr in the globular domains. This does not seem to correspond with the data shown in Fig. 3a, which appears to show more 30-31% in Ser/Gly. Please correct or clarify the difference.

37% is the Ser/Thr percentage without counting the N-terminal helix. When including the N-terminal helix and the signal peptide, it is 30%. We have clarified this in the main text.

6. In Ln 149-160 the authors describe the indications of glycosylation in the two T4P structures solved. It would be good to mention whether glycosylation is also seen in the MS fingerprint peptides, and if not, why.

Efforts were made to look for glycosylation by MS but were unsuccessful. This is not so surprising, as we had very few peptide fragments, since the pili cannot be efficiently digested. Further, we do not know what sugars are present, and the same sugar is not necessarily on each identical peptide.

7. Unless for my oversight, I did not find Suppl. Figure 3 cited in the main text, and in line 165,

“Suppl. Fig. 3” should be “Suppl. Fig. 4). Also not that in Figure 3, the caption of panels a and b does not correspond with the order of the panels in the Figure.

Thank you for pointing this out. We have fixed these errors, and put in a mention of Supp. Fig. 3.

8. In Jarrell and Albers 2012 (ref <https://doi.org/10.1016/j.tim.2012.04.007>) the name “archaellum” was proposed for the “archaeal flagellar filaments”. This reviewer is of the opinion that that proposed nomenclature does well in highlighting the fundamental difference in bacterial and archaeal flagella, and would advise the authors to adopt the nomenclature in their manuscript.

We are very well aware of the suggestion (Jarrell and Albers, 2012) to rename the archaeal flagellum as the archaellum, on the basis that it has no homology with the bacterial flagellum, as the reviewer notes. But the bacterial flagellum has no homology with the eukaryotic flagellum, so this argument is not compelling. Albers and colleagues have argued that the term flagellum is no longer used for eukaryotes, having been replaced with cilium, but this is clearly not true. Many people currently work on sperm or trypanosome flagella. Birds, bats and butterflies all have wings, and these structures are not homologous. Rather, they are the product of convergent evolution, and all share the same function. No one has proposed renaming two of them. Insects and humans have legs, but these have no homology. Similarly, sperm, trypanosomes, bacteria and archaea have flagella, and these do not reflect homology but rather convergent evolution to yield structures with similar functions. We, along with others (Wirth, 2012), thus choose to use the term archaeal flagellum in describing the tails that give these cells motility.

References

- Baquero, D.P.C., P.; Piochi, M.; Bartolucci, S.; Liu, Y.; Cvirkaite-Krupovic, V.; Prangishvili, D.; Krupovic, M. (2020). New virus isolates from Italian hydrothermal environments underscore the biogeographic pattern in archaeal virus communities. *The ISME Journal in press*.
- Jarrell, K.F., and Albers, S.V. (2012). The archaellum: an old motility structure with a new name. *Trends in microbiology* 20, 307-312.
- Kandiba, L., and Eichler, J. (2014). Archaeal S-layer glycoproteins: post-translational modification in the face of extremes. *Front Microbiol* 5, 661.
- Klingl, A., Pickl, C., and Flechsler, J. (2019). Archaeal Cell Walls. *Subcell Biochem* 92, 471-493.
- Wang, F., Cvirkaite-Krupovic, V., Kreutzberger, M.A.B., Su, Z., de Oliveira, G.A.P., Osinski, T., Sherman, N., DiMaio, F., Wall, J.S., Prangishvili, D., *et al.* (2019). An extensively glycosylated archaeal pilus survives extreme conditions. *Nat Microbiol* 4, 1401-1410.
- Wirth, R. (2012). Response to Jarrell and Albers: seven letters less does not say more. *Trends in microbiology* 20, 511-512.